# Survey and Molecular Characterization of *Echinococcus granulosus sensu stricto* from Livestock and Humans in the Altai Region of Xinjiang, China

**DOI:** 10.3390/pathogens12010134

**Published:** 2023-01-13

**Authors:** Baoping Guo, Li Zhao, Lu Zhao, Rongsheng Mi, Xu Zhang, Bingjie Wang, Gang Guo, Yuan Ren, Wenjing Qi, Zhuangzhi Zhang

**Affiliations:** 1State Key Laboratory of Pathogenesis, Prevention and Treatment of High Incidence Diseases in Central Asia, Xinjiang Medical University, The First Affiliated Hospital of Xinjiang Medical University, No.137, Liyushan Road, Urumqi 830054, China; 2Veterinary Research Institute, Xinjiang Academy of Animal Sciences, No. 726, Dongrong Street, Urumqi 830013, China; 3College of Life Sciences, Shaanxi Normal University, No. 620 Xi Chang‘an Avenue, Xi’an 710119, China; 4Key Laboratory of Animal Parasitology of Ministry of Agriculture, Laboratory of Quality and Safety Risk Assessment for Animal Products on Biohazards (Shanghai) of Ministry of Agriculture, Shanghai Veterinary Research Institute, Chinese Academy of Agricultural Sciences, Shanghai 200241, China

**Keywords:** *Echinococcus granulosus sensu stricto*, prevalence, genetic diversity, *cox1*, Altai

## Abstract

Cystic echinococcosis (CE), caused by the metacestode *Echinococcus granulosus sensu stricto* (*s*.*s*.), is an important zoonotic parasite, endemic in the Altai region of China. It is a serious human health risk and causes livestock losses. To evaluate the prevalence, genetic variation, and population structure of CE, 2898 sheep and 703 cattle were examined from October 2019 to mid-February 2020 in the Altai region (Altai, Habahe, Fuhai, and Buerjin). Sheep had an infection rate of 4.52% (131/2898) and cattle had an infection rate of 4.84% (34/703). In total, 180 cyst isolates were obtained, including 131 sheep, 34 cattle, and 15 from CE human patients. The cysts were investigated using mitochondrial cytochrome C oxidase subunit 1 (*cox1*). Polymerase Chain Reaction (PCR) results showed that, among the two genotypes of *E. granulosus s*.*s*., there were 22 different haplotypes (Haps). Phylogenetic analysis and parsimony network indicated that seventeen (77.27%) Haps belonged to the sheep strain (G1 genotype) and five Haps (22.73%) belonged to the buffalo strain (G3 genotype). Hap3 was the most common haplotype (65.00%, 112/180), which belongs to the G1 genotype. Hap18–Hap22 were found in human samples, indicating that sheep and cattle reservoirs of human CE. Molecular diversity indices revealed the high levels of haplotype diversity and relatively low levels of nucleotide diversity. Tajima’s D and Fu’s Fs tests displayed that the Altai population had a significant deviation from neutrality. Based on pairwise fixation index (Fst) values, a low level of genetic differentiation was found between the populations of *E. granulosus s*.*s*. isolated from different regions. The present survey findings represent an epidemiological survey of CE in the Altai region where there were two genotypes simultaneously and will provide more information on the genetic structure of *E. granulosus s*.*s*. within this region.

## 1. Introduction

Cystic echinococcosis (CE) is one of the most important zoonoses [1], causing a significant public-health concern and huge economic losses worldwide [2,3]. In particular, this disease causes high morbidity and mortality in Xinjiang, China [4]. Xinjiang is the major endemic area of the disease, which is caused by *Echinococcus granulosus sensu stricto* (*s*.*s*.) [5]. The Altai region is a key CE epidemic area in Xinjiang, China. Since “Control CE Action Plan (2010–2015)” [6], CE prevalence among livestock in the Altai region has been rarely reported. Molecular genetic studies, based mainly on partial sequencing of mitochondrial DNA (mtDNA), elucidate the extent of strain variation, resulting in the ‘genotype’ nominature of *E. granulosus s.s*. intraspecial variation [7,8]. Eight Genotypes of *E. granulosus sensu lato* (*s*.*l*.) have been identified in different geographical areas using mtDNA sequences [1,9]. According to “International consensus on terminology to be used in the field of echinococcoses” published in the journal *Parasite* in 2020, the new classification was *E. granulosus s*.*s*. (G1 and G3) [10,11], *E. equinus* (G4), *E. ortleppi* (G5) [11,12], *E. canadesis* (G6/7 and G8/10) [13,14,15,16], and *E. felidis* (lion strain). The G2 genotype is not a separate strain or even a monophyletic cluster but belongs to the G3 genotype [17], while the G9 genotype is no longer recognized as a distinct strain, as it is probably a microvariant of the G7 genotype [10]. The G1 genotype from sheep appears to be the dominant strain occurring in Xinjiang [6,18].

The Altai region is located in the northwest Xinjiang Uygur Autonomous Region, which is connected to Kazakhstan and Russia in the northwest and bordered by Mongolia in the northeast. It is an important basis of grassland animal husbandry in Xinjiang and it covers 147.47 million hectares of natural grassland. It is also one of the most important hydatid epidemic areas in Xinjiang. In this area, there are many cattle and sheep in the Altai region. Sheep accounted for more than 60% of the total livestock and cattle are 19% of the total. The disease infection rate of sheep, cattle, camels, and horses has been reported to be 9.8%, 8.4%, 6.8%, and 4.3%, respectively [5]. Guo et al. found that 3.5% of the sheep and 4.1% of cattle were infected with CE.

Altai city, Habahe county, Fuhai county, and Buerjin county in the Altai region have a large slaughter volume; these four counties were the main investigation points in this study. We focused on the slaughterhouses of the four areas receiving large numbers of livestock to investigate the prevalence and genetic characteristics of *E. granulosus s*.*s*. strain. This information will improve our understanding of CE in the region and will determine the relative contribution of each livestock species to the distribution and transmission of poliovirus. The study assessed the prevalence of *E. granulosus s*.*s*. in humans in the Altai region, with infection rates ranging from 0.3% to 3% [19]. However, the genetic variation in *E. granulosus s*.*s*. in the Altai region remains unclear. This paper investigates the epidemiology of *E. granulosus s*.*s*. and identifies the genetic variation in *E. granulosus s*.*s*. in the Altai area of Xinjiang. The main purpose of this study was to investigate the infection rate of domestic animal cystic echinococcosis in Altay area and to analyze the genotype of the infection. The results of the present study provide a baseline survey for further molecular anti-parasite approaches and for implementing effective local control plans in Altai. We proposed further molecular and biological studies to determine the occurrence of other genotypes/strains of *E. granulosus s.s*. in livestock and humans in order to confirm the exact source of this zoonotic infection in the Altai region, Xinjiang.

## 2. Results

### 2.1. Prevalence of CE in Livestock

In total, 3601 animals at slaughter were screened during the survey. The general prevalence of CE in the current area was 4.58% (165/3601) and included sheep at 4.52% (131/2898) and cattle at 4.84% (34/703) (Table 1). The highest prevalence of sheep was found in Habahe county (5.84%, 51/873) and the lowest prevalence of sheep was found in Buerjin county (2.58%, 11/426). There was no significant difference in the infection rate of sheep among the four regions via chi-square analysis. The highest prevalence of cattle was also found in Altai city (6.02%, 23/382) and the lowest prevalence in cattle was found in Buerjin county (2.63%, 2/76). There was no significant difference in the infection rate of cattle among the four regions via chi-square analysis. The infection rate of cattle was higher than that of sheep, but there was no significant difference in the positive rate of sheep and cattle in the four regions via Chi-square analysis (*χ*^2^ = 0.3695, *p* = 0.7192). 

### 2.2. Infection Rates by Animal Age

The age of infected sheep ranged from 10 months to more than three years old according to the degree of tooth wear. These sheep were divided into three age groups: <10 months, 10 months to 3 years, and ≥3 years with infection rates of 1.52% (23/1512), 7.62% (78/1023), and 8.26% (30/363), respectively (Table 2). Among these different age groups, the age group ≥3 years has a higher rate of infection. The percentage of positive sheep by age group was 17.56%, 59.54%, and 22.90% for weaned lambs, adult sheep, and aged sheep, respectively. These results showed that the infection rates of aged sheep (χ^2^ = 6.637, *p* < 0.001) and adult sheep (χ^2^ = 7.369, *p* < 0.001) were significantly higher than those of weaned lambs.

The cattle results were similar to those of sheep, and cattle were also divided into three age groups: <2 years, 2–5 years, and ≥5 years with infection rates of 1.94% (3/155), 4.93% (21/426), and 8.20% (10/122), respectively (Table 2). The highest infection rate among the age groups occurred at age ≥5 years. The frequency percentages of positive cattle in the age groups was 8.82%, 61.76%, and 29.41% for calves, adult cattle, and aged cattle, respectively. Statistical analysis showed that aged cattle had significantly higher infection rates than calf (χ^2^ = 2.327, *p* = 0.020), but the rate was not significantly different from that of adult cattle (χ^2^ = 1.337, *p* = 0.1684).

### 2.3. Infection Organs

In all infected livestock, the liver and lungs showed the only cysts and the liver was the most severely infected organ (Table 1). Across infected livestock, 72.73% (120/165) were from infected liver and 27.27% (45/165) were from infected lungs (Table 1). No cases of simultaneous liver and lung infection were found in sheep and cattle.

### 2.4. Genotypic Characterizations

In total, 190 cysts were subjected to PCR for amplifying the *cox1* gene fragment and these 180 samples were successfully genotyped, including 131 sheep, 34 cattle, and 15 human cysts. Compared with the publicly available *cox1* sequences (AF297617 and KJ559023), all isolates were defined as belonging to the G1 and G3 complex (Table 3). Thus, 166 (92.22%) isolates, including 124 sheep, 30 cattle, and 12 human isolates, were identified as the G1 genotype. Fourteen (7.78%) isolates, including seven sheep, four cattle, and three human isolates, were identified as having the G3 genotype (Figure 1).

### 2.5. Phylogenetic Analysis

Based on the comparison of the *cox1* gene fragment, 22 different Haps were detected and named Hap1–Hap22 (GenBank accession numbers: MW843575-MW843596) (Table 3). Seventeen (18.18%) Haps, including Hap1–4, Hap6, Hap8, Hap10–11, Hap13–18, Hap20, and Hap22, belonged to the G1 genotype. Five (22.73%) Haps, including Hap5, Hap7 Hap12, Hap19, and Hap21, belonged to the G3 genotype (Figure 1). Among all Haps, Hap3 was the most common variant, and it accounted for 117 (65.00%) isolates, including 95 sheep, 16 cattle, and 6 human isolates. In the present study, human isolates had two Haps, including Hap3, Hap5, Hap6, Hap8, Hap18, and Hap22, including twelve (80.00%) human isolates classified as G1 genotype. Hap5, Hap19, and Hap21, including three (20.00%) human isolates, were classified as G3 genotype.

### 2.6. Network Analysis

Parsimony network analysis was created to build the genealogical relationship among the 22 Haps in Table 3. Hap4 was the center of the network and 13 Haps were connected in a star-like configuration (Figure 2). Hap2, Hap4, Hap7, and Hap21 formed a diamond. Among all in the network, Hap3 (65.00%) was the major haplotype. Some groups have similar genetic distance from Hap4, Hap2, Hap7, Hap21, and Hap12. Hap14 was closely related to Hap3 and far from Hap4, and it is likely that Hap14 is a branch of Hap3. Hap15, Hap1, Hap5, and Hap18 may be sub-branches of Hap2. The network distances between Hap12 and Hap4 confirm that they belong to different genotypes. Hap14 is further from Hap13, Hap16, and Hap22, which equal two mutational steps, suggesting that their genetic relatedness was weak and indicating that Hap14, Hap9, Hap8, Hap10, Hap16, Hap4, Hap11, Hap6, Hap13, Hap17, Hap3, Hap2, Hap15, and Hap18 belong to the G1 genotype. The remaining Haps belong to the G3 genotype.

Diversity indices for different populations of *E. granulosus s*.*s*. in the Altai region were calculated using the *cox1* gene fragment. The haplotype diversity ranged from 0.438 to 0.659, while the nucleotide diversity ranged from 0.00182 to 0.00405 (Table 4). These results showed the high levels of haplotype diversity and the relatively low levels of nucleotide diversity within *E. granulosus s*.*s*. in this region. These low values suggest that the populations are not genetically differentiated. Table 5 shows values for genetic distance between subpopulations of *E. granulosus s*.*s*. sequences identified in Altai regions where samples were available. According to the Fst values, it can be considered that there is no genetic differentiation among parasite subgroups in the Altai region. Further, a low Fst value (0.01840) was found when comparing the parasite populations of humans and livestock in the Altai regions (Table 6). Finally, we analyzed the genetic distance of sheep, cattle, and humans and the result showed a low Fst value (Table 7).

## 3. Discussion

The study of interspecific genetic variation in *E. granulosus s.l.* species has made an important contribution to the study of the epidemiology, geographic distribution, and phylogeny of *E. granulosus*. Since earlier studies in the 1990s were conducted, it was not clear that there was a certain degree of variability within *E. granulosus s.l.* Subsequent studies on the genetic variability in isolates causing CE have clarified the taxonomy of the parasite, grouping a number of species under the complex *E. granulosus s.l.* The methods described by Nakao et al. [29] and Yanagida et al. [26] were used to analyze *E. granulosus s.s*., allowing us to produce data sets comparable to other sequences from different geographic regions. In total, the slaughterhouses in the four counties of the Altai region are relatively large, so they were selected for study. 

We studied the prevalence of CE in sheep (4.52%) and cattle (4.84%) in the Altai region of Xinjiang. The prevalence reported is within the range reported for Xinjiang in previous studies. These studies include the work of Meng et al. [5] (10.7% in sheep, 7.4% in cattle) and Guo et al. [6] (4.6% in sheep and 4.5% in cattle). The present study examined 2898 sheep and 703 cattle and the data suggest an endemic steady state with a reduced infection pressure in these hosts. This conclusion is consistent with the results of Guo et al. [6], who reached the same conclusion for CE prevalence in northern Xinjiang. Four selected regions had infection rates ranging from 2.58% to 5.84% in sheep and 2.78% to 6.02% in cattle, with no difference between regions. The prevalence of sheep was 4.52%, which was lower than a previous sheep study in Xinjiang (9.8%) by Meng et al. [5]. The sheep surveyed in this study were mostly less than 10 months old and, similar to a previous study, the infection rates increased as the sheep aged [5]. Since cattle are generally slaughtered after 2 years of feeding, the growth period of cattle is longer than that of sheep, which leads to a higher infection rate of cattle than that of sheep. Due to the insufficient positive sample size, the statistical difference is not significant. Through the implementation of the national prevention and control of echinococcosis (2010–2015) [30], infection from livestock in these places has also decreased somewhat. Earlier studies reported somewhat greater infection rates compared to the present study. The difference in infection rates may be attributed to sample size, unhygienic measures used at the slaughterhouse, and abiotic climatic factors that favored the onset of infection. The prevalence of parasites in slaughtered sheep and cattle is a feasible and economical way to assess the infection rates of the four regions. Although this sampling technique has the limitation that it is not a population-based random sampling survey and is unable to cover all slaughtered animals at the sites investigated, the advantages of a survey on slaughtered sheep and cattle are apparent.

We looked at the teeth of each animal to estimate its age and divided each species into groups based on age. The infection rate is generally higher in older livestock compared to younger individuals. The higher prevalence of old age may be mainly due to age-related diseases and the chronic nature of the disease [31]. The age variation can be associated with differences in exposure to infection because aged animals may have been exposed to more infective periods. Similar trends in the prevalence of CE infection in sheep and cattle clearly indicate a higher risk of disease in older livestock compared to younger ones [29,32]. 

Here, it can be seen from the results that, first of all, females are usually numerically more represented than males in ultrasound screening, which increases the possibility of gender selection bias in sampling [33], which is related to the small sample size. The relationship between sex and echinococcosis has been studied in many studies [34,35,36], but inconclusively. Secondly, our study was similar to the result of Khan et al. [36]. People in the 21–50 age group were the main labor force and had a higher probability of exposure to pathogens than other groups, and the risk of infection was correspondingly higher. In addition, the samples in this study were all from hepatic echinococcosis samples, without pulmonary echinococcosis samples. Finally, the relationship between ethnic groups and echinococcosis was not described here due to the limited sample size.

The overall results of this study clearly indicate that overall nucleotide diversity and haplotype diversity are low. Tajima’s D was negative in all analysis sequences, indicating Altai population expansion and/or purification selection. In the *cox1* sequence comparison, the negative Fu’s F value was significantly higher, indicating the presence of a rare haplotype, which could explain the expansion or hitchhiking of the *E*. *granulosus s.s*. population. The Fu’s F test was developed based on the distribution of haplotypes or alleles. A slight deviation from neutrality was recorded in the Fuhai genotypes, with low negative values but no significant deviation from neutrality was recorded in the Habahe and Buernjin genotypes (Table 4). There were significant differences between Altai and Fuhai, but no significant differences between Habahe and Buernjin, indicating that the genotypes of Altai and Fuhai are in a state of expansion that is greater than that of Habahe and Buernjin. Fst values do not support the differentiation in the subpopulation of *E. granulosus s.s.* in the Altai region (Table 5, Table 6 and Table 7). After multiple investigations of genetic variation in subpopulations (Table 5, Table 6 and Table 7) of *E. granulosus s.s.,* values remained low in variation and actually differed in pathogenicity, biological characteristics, and host responses. 

In this study, nine Haps of *E. granulosus s*.*s*. were identified in human CE cases, while five Haps were present only in humans. Nevertheless, the *cox1* gene of *E. granulosus s.s*. shows considerable genetic variation, and the Fst index of *cox1* gene from humans is obviously higher than that from livestock. That study indicated that the *E. granulosus* human subpopulations are distinct from the subpopulations of other hosts. The statistically distinct Fst values of the human *E. granulosus* isolates as compared to livestock indicate the presence of surplus variable alleles, further indicating the limited gene flow, which probably suggests that human Haps may come from sheep and cattle, which may be an infection from a definitive host infection with the same Haps.

G1 was the most common genotype found. G1 genotypes have a greater worldwide distribution than the G3 genotype [37], while the G3 genotype was the least common. In this study, the G1 genotype has 17 Haps and 166 sequences, suggesting a high rate of G1 variation. This is the first documentation that the two strains occur simultaneously in this area. Earlier studies reported that G1 and G3 genotypes of *E. granulosus s*.*s*. infect humans and livestock in Xinjiang [6]. However, we found two strains (G1 and G3) in the Altai region. The G1 genotype is the predominant epidemic *E. granulosus s*.*s*., which is similar to previous results [6,24,38] and is also responsible for the majority of human and livestock infections in many countries, such as China, Iran, Brazil, Italy, and Turkey [9]. All sequences in the present study, designated as Haps (Hap1–Hap22), form a clear clade together with the G1 and G3 genotypes expressing *E. granulosus s*.*s*. strains in the G1 and G3 complex. These results provide support for considering the G1 and G3 complex as a discrete species [39]. Both known strains (G1 and G3) of *E. granulosus s*.*s*. were found in four regions, indicating that G1 and G3 were widely distributed and had no regional differences. Hap3 (65.00%), detected in 117 isolates, was the major haplotype and showed complete sequence homology with previous reports from China [6], Turkey [40], Argentina [27], Australia [41], Iran [26], and Kyrgyzstan [22].

G1 was the most common genotype found. G1 genotypes have a greater worldwide distribution than the G3 genotype [37], while the G3 genotype was the least common. In this study, the G1 genotype has 17 Haps and 166 sequences, suggesting a high rate of G1 variation. This is the first documentation that the two strains occur simultaneously in this area. Earlier studies reported that G1 and G3 genotypes of *E. granulosus s*.*s*. infect humans and livestock in Xinjiang [6]. However, we found two strains (G1 and G3) in the Altai region. The G1 genotype is the predominant epidemic *E. granulosus s*.*s*., which is similar to previous results [6,24,38] and is also responsible for the majority of human and livestock infections in many countries, such as China, Iran, Brazil, Italy, and Turkey [9]. All sequences in the present study, designated as Haps (Hap1–Hap22), form a clear clade together with the G1 and G3 genotypes expressing *E. granulosus s*.*s*. strains in the G1 and G3 complex. These results provide support for considering the G1 and G3 complex as a discrete species [39]. Both known strains (G1 and G3) of *E. granulosus s*.*s*. were found in four regions, indicating that G1 and G3 were widely distributed and had no regional differences. Hap3 (65.00%), detected in 117 isolates, was the major haplotype and showed complete sequence homology with previous reports from China [6], Turkey [40], Argentina [27], Australia [41], Iran [26], and Kyrgyzstan [22].

The study could have epidemiological significance and it could have a direct impact on public health in the Altai region. Because of scarce data about *E. granulosus s*.*s*. in Altai, genotype identification should be the first step for controlling CE infections in this endemic area. This research could increase information about the global cycle of different genotypes of *E. granulosus s*.*s*. The confirmation of (G1–G3) complex of *E. granulosus s*.*s*. in livestock and humans in the Altai of Xinjiang reinforces the need for greater attention to the essential genetic approach for assessing all species and variants associated with echinococcosis.

## 4. Materials and Methods

### 4.1. Collection of Cysts

All the livestock that was examined for CE came from designated slaughterhouses in four areas: Altai city, Habahe county, Fuhai county, and Buerjin county (Figure 3). In total, 2898 sheep and 703 cattle were examined for cysts in their livers and lungs from October 2019 to mid-February 2020. Fifteen liver cyst isolates were also obtained from CE patients in local hospitals. Each single cyst of *E. granulosus s*.*l*. was treated as an isolate in 75% alcohol at the beginning of collection in all slaughterhouses. After returning to the experiment, protoscoleces (PSCs) and/or germinal layers were collected from cysts of livestock, washed 10 times with phosphate-buffered saline (PBS) [42], and stored at −20 °C until genomic DNA extraction.

### 4.2. CE Cases

In total, 15 human CE cases were collected in October 2018. We consulted relevant electronic medical records and found some information about the cases, such as sex, location, ethnic group, and other information (Appendix A). The results showed that the majority of the cases were female, which was twice the number of cases of males, and all cases were CE2, CE3, and CE4 cysts (Appendix A). Patients were divided into 5 periods according to different age groups [43], with most cases in the 21–50 age group (average 31.87 years old) (Appendix A). The results showed that among the 15 CE cases who were all hepatic echinococcosis, there were 7 cases of Han, 4 cases of Uyghur, and 3 cases of Mongol, respectively (Appendix A). 

### 4.3. DNA Extraction and PCR Amplification

Genomic DNA was extracted using the QIAmp DNeasy blood and Tissue Kit (Qiagen) following manufacturer instructions. Extracted DNA was stored at 4 °C for Polymerase Chain Reaction (PCR) amplification. A fragment of mitochondrial cytochrome C oxidase subunit 1 (*cox1*) (~381 bp) was amplified using the conserved forward primers: 5′-TTTTTGGGCATCCTGAGGTTTAT-3′ and reverse primers: 5′-TAAAGAAAGAACATAATGAAAATG-3′ [44], under the following conditions: initial denaturation at 94 °C for 5 min, followed by 35 cycles of 30 s denaturation at 94 °C, 30 s annealing at 55 °C, and 72 °C for 30 s, followed by a final extension at 72 °C for 7 min. The PCR reaction mixture contained: 25 µL Master Mix (GoTaq1 Green Master Mix, Promega, Madison, WI, USA), which contained 1 U Taq polymerase, 400 µM dNTP, 3 mM MgCl_2_, and 0.3 µM of each primer, 5 μL of DNA template, and nuclease-free water was added up to 50 µL PCR reaction mixture. A positive and negative control (without genomic DNA) was amplified in each PCR reaction. PCR products (10 μL) were run on 1.5% agarose gel electrophoresis to detect the single bands. After purification by the QIAquick Gel Extraction Kit (Qiagen), the PCR products were sent for sequencing to Biomed, Beijing, China.

### 4.4. Sequence Analysis

The size of the amplified DNA fragments obtained from PSCs and/or germinal layer isolates was 446 bp. From the design nucleotide sequences obtained from the isolates, unambiguous sequences of at least 381 bp were obtained for each sample. The obtained sequences were multiple aligned using Clustal Χ2 and BioEdit 5.0 and compared with available sequences of the *cox1* from the GenBank database using BLAST analysis (http://www.ncbi.nlm.nih.gov/Blast.cgi, accessed on 5 May 2020). Phylogenetic analysis was conducted using the MEGA 7.0 program. Trees were generated using neighbor-joining (NJ), maximum parsimony (MP), and maximum likelihood (ML) methods based on the Kimura 2 parameter. The phylogenetic tree was tested by bootstrapping using 1000 replicates. A network of mtDNA haplotypes (Haps) was illustrated by Network 10.1 using statistical parsimony. Population diversity indexes (number of Hap, haplotype diversity (Hd), and nucleotide diversity), Tajima’s D, and Fu’s Fs were calculated to test neutrality using the same program (DnaSPv 5.1). The genetic distance between the two subpopulations was analyzed using pairwise fixation index (Fst) and compared with Arlequin 3.5 [45].

### 4.5. Statistical Analysis

The infection rate of *E. granulosus s.s*., estimated as the percentage of positive samples with 95% confidence interval (CI), and the prevalence of four regions were determined using IBM SPSS Statistics version 22.0 (IBM Corp, Armonk, New York, NY, USA) for prevalence. Through Pearson’s chi-square test (χ^2^ test) analysis, when *p* < 0.05, the difference was considered significant.

## 5. Conclusions

This work uncovered, for the first time, detailed prevalence information regarding *E. granulosus s*.*s*. infection in livestock, and the genetic variability in *E. granulosus s*.*s*. from four different regions contributed to the taxonomy and systematic geographical knowledge of these parasites. This study showed that the G1 genotype was predominant in all genotypes of *E. granulosus s.s.* (G1, G3) and emphasized the important role of this genotype in CE distribution in livestock and human populations. Molecular diversity indices revealed high levels of haplotype diversity and relatively low levels of nucleotide diversity. Tajima’s D and Fu’s Fs tests showed that a significant deviation from neutrality is present in Altai. Fst values confirmed that a low level of genetic differentiation was found between the populations of *E. granulosus s*.*s*. isolated in the Altai regions. The findings of this survey provide valuable information on the genetic structure of *E. granulosus s*.*s*. in the Altai region.

## Figures and Tables

**Figure 1 pathogens-12-00134-f001:**
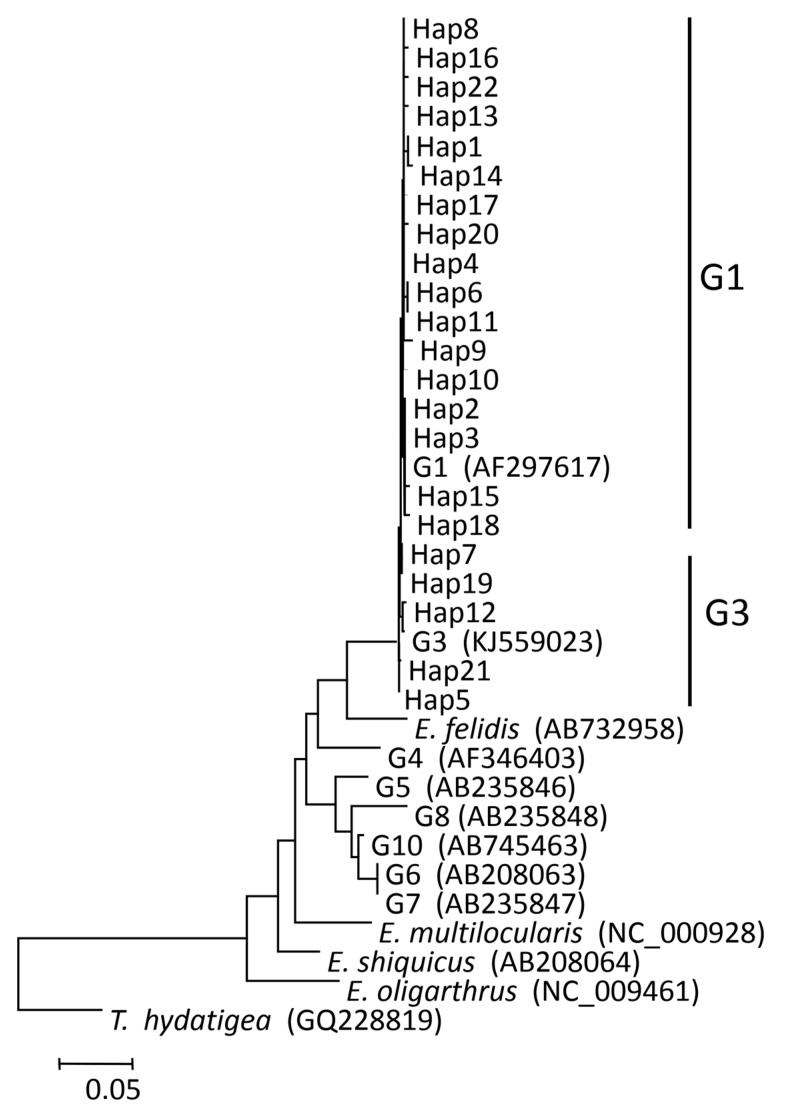
Phylogenetic trees were constructed from haplotypes (Haps) 1–22 and reference sequences retrieved from previous studies of *Echinococcus granulosus* (*s.s*.) from the Altai region (in this study). For an explanation of the colors in this graphic legend, refer to the web version of this article.

**Figure 2 pathogens-12-00134-f002:**
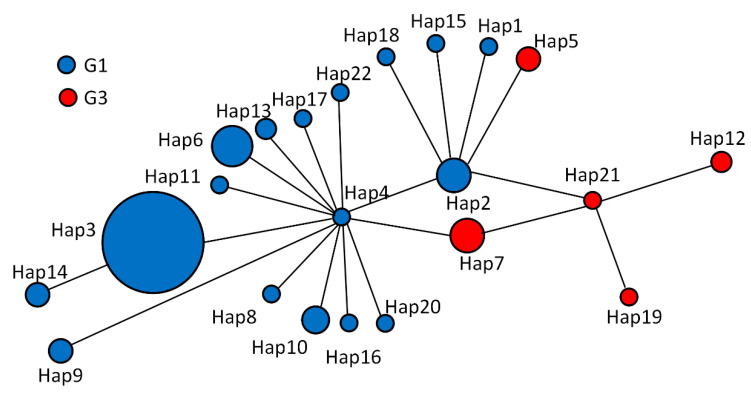
Haplotype (Haps) networks were generated from partial mitochondrial nucleotide sequences of cytochrome c oxidase subunit 1 of *Echinococcus granulosus sensu stricto* (*s.s.*) isolated from sheep, cattle, and human hydatid in Altai, Xinjiang, China. A circular icon represents haps containing the same sequence, and the circle size is proportional to the number of individuals showing such specific Haps in this study (refer to the web version of this legend for an explanation of the colors in this legend). 2.7. Population analysis.

**Figure 3 pathogens-12-00134-f003:**
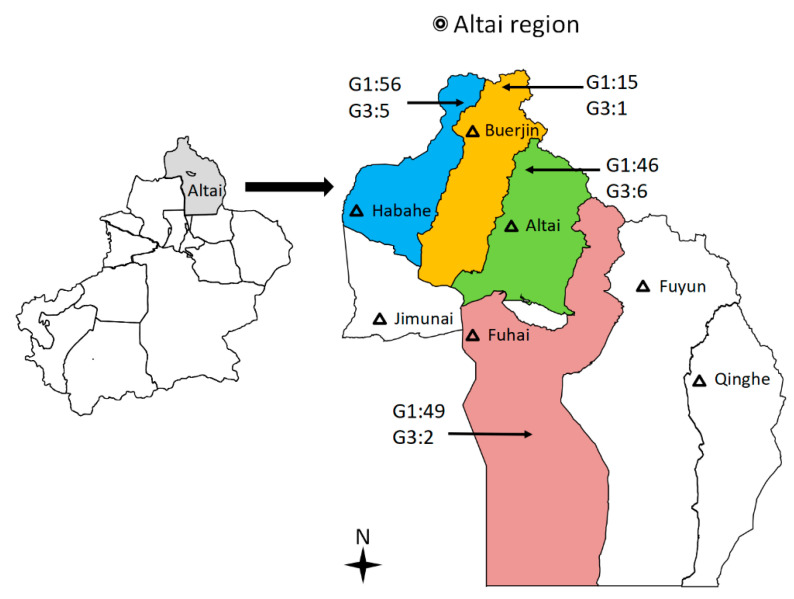
Map indicating the geographical origins of slaughterhouses in four counties of the Altai region of Xinjiang, China, from which *Echinococcus granulosus*
*sensu stricto* (*s*.*s*.) samples were collected. Color areas = cysts samples collected from four counties. Black triangle = major cities in the Altai region of Xinjiang.

**Table 1 pathogens-12-00134-t001:** Prevalence of CE in livestock from different regions of Altai.

Location	No. Samples	No. Positive			Infection Rate (%)	95% CI	Genotypes	
		Liver	Lung	Total			G1 (%)	G3 (%)
**sheep**								
Altai	625	17	8	25	4.00	2.46–5.54	21 (85.00)	4(15.00)
Habahe	873	36	15	51	5.84	4.25–7.35	47 (92.16)	4 (7.84)
Fuhai	974	34	10	44	4.52	3.20–5.80	44 (100)	0 (0)
Buerjin	426	8	3	11	2.58	1.01–3.98	14 (100)	0 (0)
Total	2898	95	36	131	4.52	3.75–5.25	123 (93.89)	8 (6.11)
**Cattle**								
Altai	382	18	5	23	602	3.62–8.38	21 (91.30)	2 (8.70)
Habahe	163	4	2	6	3.68	0.8–6.59	5 (83.33)	1 (16.67)
Fuhai	82	2	1	3	3.66	0–7.79	3 (100)	0 (0)
Buerjin	76	1	1	2	2.63	0–6.18	2 (100)	0 (0)
Total	703	25	9	34	4.84	3.25–6.43	31 (91.18)	3 (8.82)

**Table 2 pathogens-12-00134-t002:** The prevalence of CE by host age group in the Altai.

Location	Age	No. Livestock	No. Positive	Infection Rate (%)	95% CI	*χ*^2^/Value	*p* Value
**sheep**							
Weaned lambs	<10 months	1512	23	1.52	0.90–2.13	6.637	<0.001
Adult sheep	10 months–3 years	1023	78	7.62	6.00–9.25	7.369	<0.001
Aged sheep *	≥3 years	363	30	8.26	5.42–11.11		
Total		2898	131	4.52	3.76–5.28		
**Cattle**							
Calf	<2 years	155	3	1.94	0.26–4.13	2.327	<0.05
Adult Cattle	2–5 years	426	21	4.93	2.87–6.99	1.337	>0.05
Aged Cattle **	≥5 years	122	10	8.20	3.26–13.13		
Total		703	34	4.84	3.25–6.43		

* Aged sheep is compared with weaned lambs and adult sheep, ** Aged cattle is compared with calf and adult cattle.

**Table 3 pathogens-12-00134-t003:** *E. granulosus* s.s. haplotypes characterized by the partial *cox1* sequence used for comparative nucleotide sequences and phylogenetic analysis.

Haplotype	Host Origin (Number)	Region	Accession Number	Identical Sequence
Hap1	Sheep (3)	Altai (3)	MW843575	KJ628328 (99.21%) [20]
Hap2	Sheep (4), Cattle (2)	Altai (2), Habahe (1), Fuhai (1), Buerjin (2)	MW843576	MT380276 [21]
Hap3	Sheep (95), Cattle (16), Human (6)	Altai (29), Habahe (41), Fuhai (36), Buerjin (11)	MW843577	MN787529 [22]
Hap4	Sheep (1)	Altai (1)	MW843578	EU006782 (99.47%) [23]
Hap5	Sheep (2), Cattle (2), Human (1)	Altai (2), Habahe (2), Buerjin (1)	MW843579	DQ356874 [24]
Hap6	Sheep (8), Cattle (2), Human (2)	Altai (2), Habahe (8), Fuhai (1), Buerjin (1)	MW843580	MN787561 [22]
Hap7	Sheep (5)	Altai (4), Habahe (1)	MW843581	MN787550 [22]
Hap8	Sheep (1), Cattle (1), Human (1)	Altai (1), Habahe (1), Fuhai (1)	MW843582	MN787553 (99.48%) [22]
Hap9	Sheep (3)	Altai (2), Fuhai (1)	MW843583	KX227135 (99.74%) [7]
Hap10	Sheep (2), Cattle (4)	Altai (3), Habahe (1), Fuhai (2)	MW843584	MG808297 [23]
Hap11	Sheep (1)	Habahe (1)	MW843585	MT380931 (99.48%) [21]
Hap12	Cattle (2)	Habahe (2)	MW843586	AB688619 [25]
Hap13	Sheep (2)	Habahe (2)	MW843587	MN787560 (99.74%) [22]
Hap14	Sheep (1), Cattle (2)	Fuhai (3)	MW843588	MT380933 [21]
Hap15	Sheep (1), Cattle (1)	Altai (1), Fuhai (1)	MW843589	MN787554 (99.21%) [22]
Hap16	Sheep (1)	Fuhai (1)	MW843590	AB688606 (97.64%) [26]
Hap17	Sheep (1), Cattle (2)	Fuhai (2), Buerjin (1)	MW843591	MN328343 (99.74%) [22]
Hap18	Human (1)	Altai (1)	MW843592	MN787553 [22]
Hap19	Human (1)	Fuhai (1)	MW843593	MG672286 [27]
Hap20	Human (1)	Habahe (1)	MW843594	MN787549 (99.74%) [22]
Hap21	Human (1)	Fuhai (1)	MW843595	MN78746 (98.95%) [22]
Hap22	Human (1)	Altai (1)	MW843596	MH050610 [28]

**Table 4 pathogens-12-00134-t004:** Diversity and neutrality indices for different populations of *E. granulosus* s.s. from four locations in Altai, calculated from *cox1* gene fragment.

Geographical Location	*n*	h	Hd	Pi	Tajima’s D	Statistical Significance	Fu’s Fs
Altai	52 (4) ^a^	13	0.438	0.00303	−2.53507	***, *p* < 0.001	−4.700
Habahe	61 (4) ^a^	11	0.448	0.00182	−1.41172	Not significant, *p* > 0.10	−3.183
Fuhai	51 (4) ^a^	12	0.659	0.00404	−1.91824	*, *p* < 0.05	−5.172
Buernjin	16 (3) ^a^	5	0.617	0.00405	−1.29984	Not significant,$*p* > 0.10	−1.402

Abbreviations: number of isolates examined (*n*), number of Hap (h), haplotype diversity (Hd), nucleotide diversity (Pi), ^a^ Indicate human sample.

**Table 5 pathogens-12-00134-t005:** Pairwise fixation index (Fst) among *E*. *granulosus s*.*s*. subpopulations from Altai region.

Region	Altai	Habahe	Fuhai	Buernji
Altai	0.00000			
Habahe	0.01008	0.00000		
Fuhai	−0.00272	0.01238	0.00000	
Buernjin	0.01616	−0.00463	−0.00870	0.00000

Fst values close to 1 indicate extreme genetic differentiation among the four subpopulations.

**Table 6 pathogens-12-00134-t006:** Pairwise fixation index (Fst) among *E*. *granulosus s*.*s*. subpopulations from human and livestock in Altai region.

Species	Human	Livestock
Human	0.00000	
Livestock	0.01840	0.00000

Fst values close to 1 indicate extreme genetic differentiation among humans and livestock.

**Table 7 pathogens-12-00134-t007:** Pairwise fixation index (Fst) among *E. granulosus*
*s*.*s*. subpopulations from sheep, cattle, and humans in Altai region.

Species	Sheep	Cattle	Human
Sheep	0.00000		
Cattle	0.01455	0.00000	
Human	0.02684	−0.00960	0.00000

Fst values close to 1 indicate extreme genetic differentiation among human and livestock.

## Data Availability

All data generated from GenBank and published papers and supporting the findings of this study are available from Appendix A.

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
