# Peer review of "Survey and Molecular Characterization of Echinococcus granulosus sensu stricto from Livestock and Humans in the Altai Region of Xinjiang, China"

_pathogens, 2023, doi:10.3390/pathogens12010134_

Round 1
Reviewer 1 Report
The Manuscript "Epidemiological and molecular characterization of Echinococcus granulosus sensu stricto from livestock and humans in the Altai region of Xinjiang, China" submitted by Guo et al in Pathogens is well-written and provides useful information to researchers.
The following are points to be addressed before acceptance
1. Provide details of the study area
2. Add information about positive and negative controls used in PCR assays
3. You can not use the word epidemiology in the title because only a few parameters such as animal species, age included, and no detailed epidemiological parameters were collected to find out any relationships with positive cases
Author Response
Dear reviewer:
Thank you very much for reviewing our manuscript. We also greatly appreciate the reviewers for their complimentary comments and suggestions. We have carried out the reviewers suggested and revised the manuscript accordingly. Please find attached a point-by-point response to reviewer’s concerns. We hope that you find our responses satisfactory and that the manuscript is now acceptable for publication.
Provide details of the study area
Response: In western China, cattle and sheep are slaughtered at designated locations in each county, usually one in each county, so we added “designated slaughterhouses” in the article, most of which are located in county towns.
Add information about positive and negative controls used in PCR assays
Response: We are careful to modify and add information about positive and negative controls used in PCR assays in line 339-340.
You can not use the word epidemiology in the title because only a few parameters such as animal species, age included, and no detailed epidemiological parameters were collected to find out any relationships with positive cases.
Response: According to your comments, we've changed title the “Epidemiological” to “Survey”.
Mr. Zhang
Reviewer 2 Report
The research " Epidemiological and Molecular Characterization of Echinococcus granulosus sensu stricto from Livestock and Humans in the 3 Altai region of Xinjiang, China." is an exciting work, mainly held in one of the regions that had an elimination program of echinococcosis; however, the manuscript needs to change as below:
- The abstract did not summarize the research.
- the authors need to check the plagiarism percentage; usually, we should not use the same paragraphs from other manuscripts.
- Figure 2. remove the other taenia spp. from the phylogenetic tree; you need only one as an outside group.
Author Response
Response to Reviewer 2 Comments
Point 1: The abstract did not summarize the research.
Response 1: According to your comments, we are careful to modify the abstract.
Point 2: The authors need to check the plagiarism percentage; usually, we should not use the same paragraphs from other manuscripts.
Response 2: According to your comments, we have reduced the relevant duplication of content.
Point 3: Figure 2. remove the other tenia spp. from the phylogenetic tree; you need only one as an outside group.
Response 3: We are careful to modify and deleted other three tenia spp., only remained one tenia spp. in Figure 2
Reviewer 3 Report
.What is the main novelty of the present research?
.Check the spelling/grammer because there are a few errors
.line 67 _The disease infection rate of sheep, cattle, camels, and horses have been reported. should rewrite to has......
.Table 2. column 3 needs to be corrected! No.sheep should change to NO. animal because the studied animals are sheep and cattle.
.Why was the regression analysis test not used for table 2 data?
There needs to be more precise. Regarding the date information of this report Line 299 September to December 2018 is the date data, while in the abstract, it is from October 2019 to mid-February 2020!!!!! Which one is true
Author Response
Response to Reviewer 3 Comments
Point 1: W novelty of the present research?
Response 1: The main purpose of this study was to investigate the infection rate of domestic animal cystic echinococcosis in Altay area and to analyze the genotype of the infection. the results of the present study provide baseline survey for further molecular approaches to anti-parasite and for implementing effective local control plans in Altai. We proposed further molecular and biological studies to determine the occurrence of other genotypes/strains of E. granulosus s.s. in livestock and human in order to confirm the exact source of this zoonotic infection in the Altai region, Xinjiang.
Point 2:Check the spelling/grammer because there are a few errors
Response 2: According to your comments, we are careful to modify the spelling/grammer in the whole article.
Point 3:line 67 _The disease infection rate of sheep, cattle, camels, and horses have been reported. should rewrite to has......
Response 3: According to your comments, we've changed “have” to “has”.
Point 4: Table 2. column 3 needs to be corrected! No. sheep should change to NO. animal because the studied animals are sheep and cattle.
Response 4: According to your comments, we've changed “No. sheep” to “No. livestock”.
Point 5: Why was the regression analysis test not used for table 2 data?
Response 5: Thank you very much for your opinion. The result is suitable for chi-square test. Because I neglected to put the statistical results in the table, I modified the table and put the chi-square results in Table 2.
Point 6:There needs to be more precise. Regarding the date information of this report Line 299 September to December 2018 is the date data, while in the abstract, it is from October 2019 to mid-February 2020!!!!! Which one is true
Response 6: Due to my negligence, the sampling time in the article is inconsistent, and the confirmed time is “from October 2019 to mid-February 2020”.
Round 2
Reviewer 1 Report
None
Author Response
Response to Reviewer 1 Comments
Thank you very much for your help and review. Good luck
Kind regards,
Zhuangzhi Zhang
Reviewer 2 Report
In this manuscript, Guo et al. described their study of the Epidemiological and molecular characterization of Echinococcus granulosus sensu stricto from livestock and humans in the Altai region of Xinjiang, China. The molecular characterization of the Echinococcus granulosus sensu stricto genotype still needs to be clarified in the different areas. The manuscript has interesting observations, but several significant weaknesses prevent it from being accepted in the present form.
The authors should improve the English and check one more time the used sentences to avoid the plagiarism
Author Response
Response to Reviewer 1 Comments
Point 1: The authors should improve the English and check one more time the used sentences to avoid the plagiarism.
Response 1: We have made some modificaton to some sentences based on your advice in line 51-54; line 105-107; line 114-115; line 149-150; line 189-194; line 203-206; line 215; line 220; line 225; line 229-235; line 235-237; line 301-309; line 328-335; line 401-409; line 423-424.
Reviewer 3 Report
dear authors
After the first review round, I recommend inserting Response 1 as the last paragraph of the introduction.
Author Response
Response to Reviewer 3 Comments
Point 1: After the first review round, I recommend inserting Response 1 as the last paragraph of the introduction.
Response1: It has been modified according to your comments in line86-line92.